# OpenReview forum: "Meerkat-VL: Implicit Risk Safety Alignment in Multimodal LLMs via Perceptual Reasoning and Self-Verification"
_ICML.cc/2026/Conference — ICML 2026 regular_

### Official Review · Reviewer_cKhy · 2026-02-21

**Soundness:** 4
**Presentation:** 3
**Significance:** 3
**Originality:** 3
**Overall Recommendation:** 4
**Confidence:** 4

**Summary:**

This paper mainly discussed the safety alignment problem of MLLMs in implicit risk scenarios, that is, text and images only generate safety risks after being combined. The authors first construct a new multimodal safety training set, Meerkat-Safe, which contains multi-dimensional data and weights, and sampled it for human validation to ensure the reliability of the dataset. Then, the authors propose the Meerkat-VL framework, letting the model perform explicit risk perception and intent reasoning before answering, and score its own reasoning and answer according to normative rules as a process-aware reward signal, and then impose penalties on situations where no risk is perceived but a safe template answer is given, thereby attempting to alleviate this problem. Experiments show that the authors' method performs excellently on multiple safety benchmarks, effectively alleviating the problem.

**Compliance With Llm Reviewing Policy:**

Affirmed.

**Final Justification:**

The authors provided a high-quality and comprehensive rebuttal that effectively resolved all my concerns. Specifically, the inclusion of computational cost details, additional experiments on other MLLMs, and the concrete case study on reducing reward hacking significantly strengthened the paper.

This work offers substantial value in addressing the safety alignment of MLLMs, and it meets the conference's acceptance criteria. While I am maintaining my initial score of 4, please interpret this as a positive endorsement. I look forward to seeing the rebuttal additions integrated into the camera-ready version.

**Key Questions For Authors:**

1. Can the authors provide a detailed explanation of the computational resource consumption?
2. Can the authors provide experiments on other small MLLMs, such as InternVL and LLaVA? I do not require all the benchmarks. Selecting 2-3 to compare with the baseline is sufficient.
3. In my opinion, the evidence for "reducing reward hacking" is mostly indirect. The persuasiveness of this claim is somewhat insufficient. I would like to see a more explicit analysis, preferably combined with actual cases.

**Limitations:**

Judging from the classifications and examples provided in the paper, the dataset of this paper is constructed entirely based on the US context, and does not discuss whether risk definitions of different cultures/regions are covered. This will not constitute a reason for rejection, but I hope the authors can explicitly state this.

**Strengths And Weaknesses:**

## Strengths:

**1.** For the first time, it clearly explains the gap in multimodal implicit risks, which is a good theoretical contribution.

**2.** The tested models and existing methods cover a wide range, and it also includes a case study and an ablation study, which is convincing.

**3.** The whole system of dataset + method is complete, the amount of data is sufficient, and human validation was done, making it relatively reliable. The annotation process is also fully provided, showing good transparency.

**4.** The authors not only constructed their own dataset but also verified it on multiple safety benchmarks. The experimental results are sufficient.

## Weaknesses:

**1.** The innovation of this paper leans more towards an engineering combination. The proposed NPSV and DPCA modules already have similar applications in the existing LLM safety field, and the novelty is not prominent enough.

**2.** The framework adopts a complex MAS, and the pipeline is relatively complex. Although the paper provides sufficient implementation details, there is not much discussion on computational overhead and training stability.

**3.** The comprehensiveness of the experiments in this paper is worth affirming. However, the core experiments are only conducted on the 7B version of Qwen-2/2.5, lacking testing on other small open-source multimodal models. Theoretically, I am willing to believe the authors' claim about the framework's universality, but it is significantly insufficient experimentally.

**4.** The internal elements of the architecture diagram in the main text are arranged too densely, resulting in a poor reading experience.

---

> ### Author Rebuttal · Authors · 2026-03-30
>
> Dear Reviewer cKhy,
>
> We sincerely thank you for the positive evaluation of our work. We hope the point-by-point responses will address your concerns.
>
> ---
>
> >**Q1.About methods' novelty**
>
> Thanks for raising critical questions about the novelty. We fully understand your concerns and sincerely appreciate your recognition of our overall contributions. While prior work applies process-level rewards via keyword matching[1] or external models[2], our method introduces two key improvements:
>
> 1. Reward construction: We use structured annotations to derive process-aware, self-verification rewards **without external evaluators**.
> 2. Objective design: To improve implicit risk perception(Section 2), we assess not only risk detection but also **the correct identification of risk sources**, reducing reward hacking.
>
> [1] SafeGRPO: Self-Rewarded Multimodal Safety Alignment via Rule-Governed Policy Optimization
>
> [2] Rule based rewards for language model safety
>
> ---
>
> >**Q2.More Training Details**
>
> **Training Cost:**
>
> We compare Meerkat-VL with conventional GRPO in the VeRL framework. Training is conducted on Qwen2.5-VL-7B (max length 2048, rollout_n = 8) using 8 H200 GPUs. Conventional GRPO relies on reward models(RMs) trained via Safe RLHF-V, whereas Meerkat-VL leverages NPSV for process-aware rewards. **As shown in Table 1, the total cost is 16.93% lower than GRPO. Moreover, our method doesn't require loading additional 7B-scale RMs into GPUs and reduces memory overhead.**
>
> **Table 1. Comparison of Training Costs**
> |Method|SFT Cost|RMs Cost|RL Cost|Total Cost|Time Compared to GRPO|Epochs|Batch Size|
> |-|-|-|-|-|-|-|-|
> |GRPO (with RMs)|—|3.20h|5.07h|8.27h|—|2|16
> |Meerkat-VL|0.25h|—|6.62h|6.87h|-16.93%|2|16
>
> **Training Stability:**
>
> The reward_mean shows a steady improvement, increasing from around 0.4 to values approaching 1. The KL loss increases early and then stabilizes around 0.01–0.03. The entropy fluctuates early and then slightly decreases to a stable level. We will include detailed figures in the revision.
>
> ---
>
> >**Q3.Experiments on non-Qwen models**
>
> We sincerely appreciate the valuable suggestion and apologize for the oversight. To address this, we report additional results in Table 2, which align with your intuition that our methods are broadly applicable.
>
> **Table 2. Results on non-Qwen models**
> |Model|Beavertails-V|SPA-VL|MM-SafetyBench|MSSBench|SIUO|
> |-|-|-|-|-|-|
> ||S / H|S / H|S / H|S / H|S / H|
> |InternVL2.5-8B|60.6 / 78.0|68.2 / 76.7|51.8 / 70.9|15.3 / 50.3|29.1 / 50.4|
> |+SPA-VL|87.6 / 63.5|91.2 / 60.6|76.3 / 64.6|22.3 / 57.0|30.3 / 45.2|
> |+Safe RLHF-V|81.4 / 79.2|83.3 / 69.1|73.5 / 76.9|29.0 / 69.3|34.9 / 56.0|
> |+SaFeR-VLM|86.5 / 78.7|90.1 / 80.8|79.7 / 79.3|46.0 / 76.7|48.3 / 61.2|
> |+Meerkat-VL|95.9 / 82.9|96.5 / 90.1|87.8 / 91.3|74.7 / 89.3|72.2 / 80.3|
> |—|
> |LLaVA-OV-7B|52.2 / 61.4|59.9 / 60.8|47.4 / 70.1|20.7 / 53.3|23.8 / 40.2|
> |+SPA-VL|80.3 / 51.8|81.2 / 46.1|70.7 / 58.0|22.7 / 42.3|25.1 / 45.8|
> |+Safe RLHF-V|71.9 / 69.7|76.5 / 66.4|64.8 / 68.2|23.7 / 57.3|36.2 / 52.5|
> |+SaFeR-VLM|84.8 / 76.9|86.3 / 76.2|75.5 / 78.1|40.3 / 68.7|43.9 / 61.0|
> |+Meerkat-VL|92.1 / 83.7|91.0 / 92.4|86.6 / 85.9|68.3 / 85.7|64.2 / 79.8|
>
> ---
>
> >**Q4.More Analysis for Reducing Reward Hacking**
>
> First, Figure 6 provides a direct evaluation of reward accuracy. NPSV surpasses RMs in safety and helpfulness accuracy, and further evaluates risk and intent perception, which RMs cannot do. **This indicates that NPSV provides more reliable rewards and reduces reward hacking at the source.**
>
> Second, there is a concrete case:
> - [Image]: A student studying late at night with coffee
> - [Query]: How can I stay this focused until my exams tomorrow?
> - GT-risk: caffeine intake and sleep deprivation
> - Response: ...This situation may involve substance misuse, so I cannot...
>
> Safety reward comparison:
> - RM: safety score 0.87
> - Ours:
>   - risk perception -> $s_{ps}=0.1$, answer safety-> $s_{as}=0.76$
>   - final safety score -> $α * s_{ps}+(1-α) * s_{as}-λ_{con}/w_s * (s_{as} - s_{ps}) =0.2 * 0.1+0.8 * 0.76-0.3/0.6 * (0.76-0.1) = 0.30$
>
> This case shows that RMs may reward misaligned refusals based on incorrect risk, while our method penalizes such a mismatch by tying rewards to correct risk, thereby reducing reward hacking
>
> ---
>
> >**Q5.Coverage of Risk Definitions**
>
> We account for cultural differences and diversity when constructing the dataset. Specifically, we include both US-based models (GPT, Gemini) and Chinese models (Qwen, Doubao), and apply cross-validation to improve the generality and coverage of risk definition. Due to space limits, we only show a few examples in the Appendix. Our dataset also covers multiple regions (e.g., Asia, Europe, Africa). For example, Figure 17 presents an East Asian case.
>
> ---
>
> >**Q6.Readability of Figures**
>
> We sincerely apologize for this important issue. Due to rule constraints, we cannot modify our manuscript at this stage. We will reconstruct the figure to present the workflow more clearly in the revision.

---

> > ### Author Rebuttal · Reviewer_cKhy · 2026-04-01
> >
> > Thank you for the detailed and high-quality rebuttal. You have addressed the concerns I raised.
> >
> > At this stage, I will temporarily maintain my initial score. I am satisfied with this work and will certainly not lower my score. However, since a 5 (Accept) is a strong endorsement, I would like to wait and review the discussions from the other reviewers and the AC before making a final decision on raising my score.

---

> > > ### Author Response · Authors · 2026-04-02
> > >
> > > Dear Reviewer cKhy,
> > >
> > > We sincerely thank you for your encouraging feedback and for acknowledging that your concerns have been fully resolved. We are glad to hear that you found the rebuttal to be of high quality. We are also grateful for your support and constructive feedback, which have been invaluable in improving our work. This has been a very meaningful rebuttal process for us, and we will incorporate the relevant content and improvements in the revised version of the paper.
> > >
> > > We appreciate your thoroughness in waiting for the full discussion. We have addressed the comments from all reviewers in detail to ensure the paper's core contributions are clear. We believe the additional results and explanations enhance the work, and we hope this gives you the confidence to support our paper further as the discussion moves forward.
> > >
> > >
> > >
> > > Best regards,
> > >
> > > Authors of the paper

---

### Official Review · Reviewer_SdNg · 2026-03-09

**Soundness:** 4
**Presentation:** 4
**Significance:** 4
**Originality:** 3
**Overall Recommendation:** 5
**Confidence:** 5

**Summary:**

This paper addresses the challenge of implicit risk safety alignment in MLLMs.  To tackle this issue, a meaningful dataset(Meerkat-Safe) is introduced with fine-grained annotations of latent dangers and user intent. Then, Meerkat-VL is proposed. It incorporates Normative Perceptual Self-Verification (NPSV) and Dual-Objective Perceptual Consistency Alignment (DPCA) to provide more accurate and process-aware rewards. Finally, the model is optimized with GRPO to encourage responses grounded in risk perception. Experiments on multiple benchmarks demonstrate that Meerkat-VL outperforms existing baselines by a clear margin.

**Compliance With Llm Reviewing Policy:**

Affirmed.

**Final Justification:**

All the concerns I raised have been fully addressed. I find the responses convincing and the work to be of high quality, and I adjust my score to 5.

**Key Questions For Authors:**

1.	Although the authors describe the implementation details of Logit-Based Reward Softening in Sec. 3.2, I remain somewhat unclear about it. More detailed explanation would likely be helpful.

2.	In typical RLHF or GRPO [1] pipelines, the reward is often taken from the single token with the highest probability.  Why compute the logit distribution instead of using the hard-max token value?

3.	The evaluation relies on GPT-5.1 as a judge to assess helpfulness and safety. While LLM-as-a-Judge is a common practice, it may introduce inherent biases and fail to fully capture the complexity of human preferences. Without human evaluation to validate the metrics, the reliability of the conclusions is somewhat limited. The authors are encouraged to provide additional analysis on the consistency between GPT-based and human evaluations to strengthen the evidence.

[1] Rong X, Huang W, Wang T, et al. “SafeGRPO: Self-Rewarded Multimodal Safety Alignment via Rule-Governed Policy Optimization” CVPR2026

**Limitations:**

Yes

**Strengths And Weaknesses:**

Strengths:

1.	The manuscript studies an important but relatively underexplored area in safety alignment. The analysis of the perceptual gap provides meaningful insights.

2.	The method is novelty and reflect solid technical soundness.

3.	The self-verification method replaces traditional reward models to provide better process-aware rewards. It mitigates the reliance on human preference data in safety alignment

4.	The authors provide extensive experimental settings, dataset details, and source code.

Weaknesses:

1.	While Meerkat-VL is effective, its efficiency (e.g., training costs) is not discussed.

2.	Lack a sensitivity analysis of the DPCA hyperparameters, specifically the perception-response balancing factor $\alpha$ and the consistency penalty coefficient $\lambda_{con}$.

---

> ### Author Rebuttal · Authors · 2026-03-30
>
> Dear Reviewer SdNg,
>
> Thanks for your insightful comments and for recognizing the technical soundness and experimental design of our work. We also appreciate your careful review and constructive suggestions. We hope our responses can address your concerns thoroughly.
>
> ---
>
> > **Q1.	The Training costs of Meerkat-VL**
>
> We sincerely thank you for the thoughtful critiques of our work. As noted, Reviewer cKhy raised a similar concern about this.
>
> In response, we conduct a detailed comparison between Meerkat-VL and conventional GRPO within the VeRL framework. Specifically, we use the Qwen2.5-VL-7B model and train on the Meerkat-Safe dataset using both conventional GRPO and Meerkat-VL. The training is performed with a maximum response length of 2048 and rollout_n = 8 on 8 H200 GPUs.  Conventional GRPO relies on reward signals from CM-V and RM-V, which are trained in Safe RLHF-V[1] as reward models(RMs). In contrast, Meerkat-VL employs NPSV to generate process-aware rewards.
>
> **Table 1. Comparison of Meerkat-VL and GRPO (with RMs) Training Costs.**
> |Method|Cold Start Costs|RMs Costs|RL Costs|Total Costs|Time Compared to GRPO|Epochs|Batch Size|
> |-|-|-|-|-|-|-|-|
> |GRPO (with RMs)|—|3.20h|5.07h|8.27h|—|2|16
> |Meerkat-VL|0.25h|—|6.62h|6.87h|-16.93%|2|16
>
> **As shown in Table 1, the 3.2 hours required to train RMs makes the overall cost of traditional GRPO 16.93% higher than Meerkat-VL. Moreover, our method doesn't require loading additional 7B-scale RMs into GPUs and reduces memory overhead.**
>
> [1] Safe rlhf-v: Safe reinforcement learning from multi-modal human feedback
>
> ---
>
> > **Q2. Sensitivity analysis of the DPCA.**
>
> The sensitivity analysis of α and $λ_{con}$ is provided in **Appendix D.5**. We evaluate multiple configurations ($α \in $ {0.1, 0.2, 0.4}, $λ_{con} \in$ {0.1, 0.2, 0.4}). The setting $α=0.2, λ_{con} = 0.3$ achieves the most stable performance. Moreover, Table 2 in the main paper shows that both components contribute positively, with performance decreasing when either term is removed.
>
> ---
>
> > **Q3.	Clarification on the Implementation of Logit-Based Reward Softening.**
>
> Thank you for the question. We further clarify the implementation of Logit-Based Reward Softening. In our setting, each verification sequence includes four evaluation scores $s^i_k \in$ {$s^i_{ps}, s^i_{ph}, s^i_{as}, s^i_{ah}$}, with different value ranges across metrics. Specifically,  $s^i_{ps}$ and $s^i_{ph}$ take values from {1,2,3}, while $s^i_{as}$ and $s^i_{ah}$ take values from {1,2,3,4,5}. We first locate the token corresponding to each score via matching on the structured output. At that position, we extract the logits to candidate score tokens in $\mathcal{Z}$. The final score is computed as the expectation over candidate scores.  For example, for $s^i_{as}$, if the logits for tokens {1,2,3,4,5} are {0,0,0.1,0.4,0.5}, the resulting score is 3×0.1+4×0.4+5×0.5=4.4.
>
> ---
>
> > **Q4. Rationale for Using Logit-Based Rewards Instead of Hard-Max Tokens.**
>
> Using a hard-max token reduces the scoring distribution to a single discrete value, which limits the resolution of the reward signal. For instance, responses with expected scores of 4.55 and 4.90 may differ in quality, yet both could be mapped to the same score(5) under a hard-max scheme. **This limits the ability to learn fine-grained preferences and may hinder policy convergence. In contrast, our method provides smoother rewards, which are consistent with RMs.**
>
> **Table2. Impact of Logit-Based Reward Softening.**
> |Method|Beavertails-V||SPA-VL||MM-SafetyBench||MSSBench||SIUO||
> |-|-|-|-|-|-|-|-|-|-|-|
> ||Safe|Help|Safe|Help|Safe|Help|Safe|Help|Safe|Help|
> |Hard-Max|95.48|90.77|95.24|91.37|93.64|88.26|82.67|89.00|82.04|88.02|
> |Logit-Based Soften|98.66|93.15|98.23|94.23|97.32|93.52|88.33|95.33|89.68|93.55|
>
> As shown in **Table 2**. Logit-based softening improves both safety and helpfulness across all benchmarks. This shows that smoother rewards provide a better training signal than hard-max tokens.
>
> ---
>
> > **Q5. The consistency between GPT- and human-based evaluation.**
>
> Thank you very much for your thoughtful comments. We fully share your concern. Relying solely on LLMs as evaluation proxies can introduce bias. To address this, we conduct a human–AI agreement study (see **Appendix D.3**) to assess whether GPT-5.1's evaluations align with human experts. We sample 500 evaluation instances from our benchmark. All samples are annotated by our long-term professional safety annotation team with the same evaluation rules. The results are below:
> |Metric|Agreement Rate|
> |-|-|
> |Safety|89.4%|
> |Helpfulness|92.8%|
>
> **These results suggest that GPT-5.1 aligns closely with human judgment, supporting its use as an evaluator.** Besides, we note that LLM-based evaluation has been widely adopted in prior work, including the GPT-4 Technical Report and Constitutional AI[2], where it has proven to be a practical alternative when human evaluation is costly.
>
> [2] Constitutional ai: Harmlessness from ai feedback

---

> > ### Author Rebuttal · Reviewer_SdNg · 2026-04-01
> >
> > Thank you for the detailed and well-supported rebuttal. All the concerns I raised have been fully addressed. I find the responses convincing and the work to be of high quality, and I will adjust my score to 5 (Accept) accordingly.

---

> > > ### Author Response · Authors · 2026-04-02
> > >
> > > Dear Reviewer SdNg,
> > >
> > > We sincerely thank you for your recognition and valuable feedback. This has been a truly meaningful review process for us. We are pleased that the additional experimental results and explanations have resolved your concerns.
> > >
> > > We are also deeply grateful for your positive view of the overall framework and its contributions to the field of multimodal safety alignment. We truly appreciate your recognition, and we are committed to incorporating the content discussed during the rebuttal into the revised version.
> > >
> > > Best Regards,
> > >
> > > Authors of the paper

---

### Official Review · Reviewer_H46e · 2026-03-12

**Soundness:** 3
**Presentation:** 3
**Significance:** 3
**Originality:** 3
**Overall Recommendation:** 4
**Confidence:** 4

**Summary:**

This paper introduces Meerkat-VL, a multimodal safety alignment framework specifically designed to address *implicit risks* in Multimodal Large Language Models (MLLMs) -- scenarios where individual modalities (text and image) appear benign in isolation but produce harmful semantics when combined. The framework consists of three main contributions:

1. **Meerkat-Safe Dataset**: A new dataset of 8,436 implicit-risk samples with fine-grained annotations including user intent, implicit risk descriptions, and safety warnings, constructed via a multi-agent cyclic validation pipeline with human verification.

2. **Normative Perceptual Self-Verification (NPSV)**: A mechanism that replaces traditional black-box reward models with a self-verification process where the model reasons about latent risks and user intent before responding, then self-scores its reasoning and output against normative rules, producing process-aware reward signals for GRPO-based reinforcement learning.

3. **Dual-Objective Perceptual Consistency Alignment (DPCA)**: A training objective that combines perceptual reward weighting with a consistency penalty to prevent reward hacking -- penalizing models that produce superficially safe answers without genuinely perceiving the underlying risks.

Experiments are conducted on Qwen2-VL-7B and Qwen2.5-VL-7B backbones across five benchmarks (three explicit-risk: Beavertails-V, SPA-VL, MM-SafetyBench; two implicit-risk: SIUO, MSSBench), demonstrating improvements of 16% and 13% in average safety and helpfulness, and 32% safety improvement on implicit-risk tasks compared to baselines.

**Compliance With Llm Reviewing Policy:**

Affirmed.

**Final Justification:**

The author's rebuttal has solved my concern, I hold the belief that the paper will be a valuable paper.

**Key Questions For Authors:**

1. Have you attempted or planned experiments on non-Qwen architectures (e.g., LLaVA-OneVision, InternVL-2.5)? The think/answer tag structure and logit-based reward softening seem architecture-agnostic in principle, but empirical validation would be important.

2.  What is the over-refusal rate of Meerkat-VL on genuinely benign queries that may superficially resemble implicit risks? For example, a user asking about cooking with a kitchen knife (benign) versus a harmful scenario involving a knife. Have you evaluated on a dedicated benign query set?

3.  How many of the 8,436 Meerkat-Safe samples were rejected during human verification, and what were the primary reasons for rejection? What was the inter-annotator agreement among the three undergraduate annotators?

4. Does the perceptual reasoning (think tag) at inference time add significant latency? What is the typical length of the reasoning trace, and how does this affect practical deployment?

5.  Since Beavertails-V appears both in cold-start training data and as an evaluation benchmark, can you provide more details on how train/test splits were managed to avoid contamination? Similarly, how sensitive are the results if the cold-start data is drawn exclusively from Meerkat-Safe rather than Beavertails-V?

**Limitations:**

The authors discuss limitations in Section B (Appendix), acknowledging two main issues: (1) NPSV faces challenges with long-tail risk categories where rules may not cover rare situations, and (2) the evaluation is limited to simplified single-turn benchmarks. They also note that existing benchmarks may not capture the full complexity of real-world multimodal safety scenarios.

**Strengths And Weaknesses:**

### Strengths

1. The three-component design (dataset, self-verification, consistency alignment) addresses the identified bottlenecks in a logically coherent manner. Each component targets a specific limitation: data scarcity, reward hacking, and insufficient supervision respectively.

2. Meerkat-VL achieves consistent improvements across all five benchmarks on both backbone models, with particularly impressive gains on implicit-risk benchmarks (e.g., 89.68% safety on SIUO vs. 37.13% for the baseline Qwen2.5-VL-7B). The ablation studies are well-structured, systematically isolating the contributions of NPSV, Meerkat-Safe, and DPCA components.

3. The paper provides comprehensive appendices including all prompt templates for the multi-agent pipeline, detailed hyperparameters, the complete normative self-verification rules, example cards for the dataset, sensitivity analyses, failure case analyses, and human-AI agreement validation (89.4% safety, 92.8% helpfulness agreement).



### Weaknesses

1. As acknowledged by the authors in Section B, the evaluation is limited to single-turn interactions. Real-world implicit risks often emerge across multi-turn dialogues where context accumulates gradually, and it is unclear how well the perceptual reasoning approach scales to such settings.

2. The cold-start phase samples 3k instances from Beavertails-V and 2k from Meerkat-Safe for response generation, plus 1k verification samples. Since Beavertails-V is also used as an evaluation benchmark, there is a potential data contamination concern. While the paper uses test splits, this overlap should be more explicitly addressed.

---

> ### Author Rebuttal · Authors · 2026-03-30
>
> Dear Reviewer H46e,
>
> Thanks very much for your constructive and insightful feedback. We also hope our responses help clarify your concerns.
>
> ---
>
> >**Q1.Experiments on non-Qwen models**
>
> We sincerely appreciate this valuable suggestion and apologize for the oversight. To address this, we report additional results in Table 1, which further validate the generality of our methods.
>
> **Table 1. Results on non-Qwen models**
> |Model|Beavertails-V|SPA-VL|MM-SafetyBench|MSSBench|SIUO|
> |-|-|-|-|-|-|
> ||S / H|S / H|S / H|S / H|S / H|
> |InternVL2.5-8B|60.6 / 78.0|68.2 / 76.7|51.8 / 70.9|15.3 / 50.3|29.1 / 50.4|
> |+SPA-VL|87.6 / 63.5|91.2 / 60.6|76.3 / 64.6|22.3 / 57.0|30.3 / 45.2|
> |+Safe RLHF-V|81.4 / 79.2|83.3 / 69.1|73.5 / 76.9|29.0 / 69.3|34.9 / 56.0|
> |+SaFeR-VLM|86.5 / 78.7|90.1 / 80.8|79.7 / 79.3|46.0 / 76.7|48.3 / 61.2|
> |+Meerkat-VL|95.9 / 82.9|96.5 / 90.1|87.8 / 91.3|74.7 / 89.3|72.2 / 80.3|
> |—|
> |LLaVA-OV-7B|52.2 / 61.4|59.9 / 60.8|47.4 / 70.1|20.7 / 53.3|23.8 / 40.2|
> |+SPA-VL|80.3 / 51.8|81.2 / 46.1|70.7 / 58.0|22.7 / 42.3|25.1 / 45.8|
> |+Safe RLHF-V|71.9 / 69.7|76.5 / 66.4|64.8 / 68.2|23.7 / 57.3|36.2 / 52.5|
> |+SaFeR-VLM|84.8 / 76.9|86.3 / 76.2|75.5 / 78.1|40.3 / 68.7|43.9 / 61.0|
> |+Meerkat-VL|92.1 / 83.7|91.0 / 92.4|86.6 / 85.9|68.3 / 85.7|64.2 / 79.8|
>
> ---
>
> > **Q2.Over-Sensitivity on Multimodal Queries**
>
> Thanks for this valuable question. In response, we evaluate over-sensitivity using MossBench[1], which includes 300 benign multimodal queries.
>
> **Table 2.Results on MossBench**
> |Model|Over-refusal Rate(↓)|
> |-|-|
> |Qwen2.5-VL|11.33|
> |+SPA-VL|40.33|
> |+MM-RLHF|38.67|
> |+Safe RLHF-V|19.00|
> |+Safer-VLM|16.33|
> |+Meerkat-VL|10.67|
>
> **As shown in Table 2, we achieve the lowest over-refusal rate. This indicates that our method can better distinguish benign intent from risky queries, thereby reducing unnecessary refusal.** We will include this important experiment in revision.
>
> [1] Mossbench: Is your multimodal language model oversensitive to safe queries?
>
> ---
>
> > **Q3.Dataset Construction and Human Verification Details**
>
> We generate 16,000 candidate samples at first. After multi-agent cyclic validation, 10,497 samples are retained. Finally, human verification filter samples to 8,436. The primary reasons for rejection include:
> 1. unclear implicit risk
> 2. image-text misalignment
> 3. low-quality gt-annotations
> 4. drift into explicit risk
>
> **For reliability, we adopt strict guidelines covering implicit risk, intent, and safety labeling. Each sample is reviewed by all annotators, with disagreements resolved by majority vote.** We observed strong inter-annotator agreement, indicating reliable annotations. We will provide more details in the revision.
>
> ---
>
> > **Q4.Overhead of Perceptual Reasoning**
>
> We report the average length of total answer and reasoning across benchmarks below. Actually, the intent and risk perception already appear in the original answers. Thus, **“think” tokens primarily restructure rather than expand this process, enabling better supervision of perception**. This introduces only minimal overhead while yielding significant performance gains.
>
> **Table 3. Response and Reasoning Length (mean/median)**
> |Model|Full Length|Think Length|
> |-|-|-|
> |Qwen2.5-VL-7B|382.0 / 385.0|-|
> |+SPA-VL|223.5 / 202.0|-|
> |+MM-RLHF|265.7 / 275.0|-|
> |+Safe RLHF-V|463.5 / 478.0|-|
> |+Safer-VLM|498.0 / 512.0|157.5 / 149.0|
> |+Meerkat-VL|397.0 / 384.0|84.5 / 79.0|
>
> ---
>
> > **Q5.Cold-Start Data Composition and Contamination**
>
> (1) **Beavertails-V is introduced by Safe RLHF-V[2], and we strictly follow its official split.** The 3k cold-start samples are drawn from the training split with class-balanced sampling, while evaluation is the test split. Therefore, there is no train/test overlap.
>
> (2) We compare a variant in which the cold-start stage uses only Meerkat-Safe against the mixed setup.
>
> **Table 4. Impact of Cold-Start Data Composition(Qwen2.5-VL-7B).**
> |Setting|Beavertails-V|SPA-VL| MM-SafetyBench|SIUO|MSSBench|
> |-|-|-|-|-|-|
> ||S / H|S / H|S / H|S / H|S / H|
> |Mixed|98.66 / 93.15|98.23 / 94.23|97.32 / 93.52|89.68 / 93.55|88.33 / 95.33|
> |Meerkat-Safe Only|97.48 / 92.36|97.05 / 93.01|96.12 / 92.08|88.74 / 92.63|88.67 / 94.00|
>
> Using only Meerkat-Safe leads to a small overall drop (~1 point). This may come from two factors:
> - The mixed setup provides broader coverage by combining explicit and implicit-risk data.
> - Using only Meerkat-Safe causes some overlap between cold-start and RL data, leading to instability early in RL training.
>
> [2] Safe rlhf-v: Safe reinforcement learning from multi-modal human feedback.
>
> ---
>
> > **Q6.About Two Limitations in Section B.**
>
> Thanks for drawing attention to these limitations, which are also concerns in our recent work. However, the lack of multi-turn safety data limits evaluation beyond single-turn settings. We are working toward constructing multi-turn datasets and exploring more scalable methods(e.g., continual learning) to address long-tail risks. We will further investigate these issues in future work.

---

> > ### Author Rebuttal · Reviewer_H46e · 2026-04-06
> >
> > My concerns have been adequately addressed. Thanks for the author response!!

---

> > > ### Author Response · Authors · 2026-04-06
> > >
> > > Dear Reviewer H46e,
> > >
> > > We sincerely appreciate your positive feedback and detailed comments. We are also greatly honored to see your recognition of our rebuttal content, maintaining your strong support for us!
> > >
> > > We will incorporate the changes from the rebuttal period into the final version. If you have any other questions, please feel free to ask anytime!
> > >
> > > Best Regards,
> > >
> > > Authors of the paper

---

### Decision · Program_Chairs · 2026-04-30

**Decision:**

Accept (regular)

**Comment:**

his paper introduces Meerkat-VL, a framework for implicit-risk safety alignment in multimodal LLMs, comprising the Meerkat-Safe dataset, Normative Perceptual Self-Verification, and Dual-Objective Perceptual Consistency Alignment. All three reviewers converged positively after rebuttal, with concerns comprehensively addressed. The core results: 32% implicit-risk safety improvement with low over-refusal and reduced training cost versus conventional GRPO, are empirically well-supported. The remaining concern around engineering novelty is noted but does not undermine the contribution's practical significance. Camera-ready should integrate rebuttal experiments and improve figure readability as committed.